# Correlation between MOVA3D, a Monocular Movement Analysis System, and Qualisys Track Manager (QTM) during Lower Limb Movements in Healthy Adults: A Preliminary Study

**DOI:** 10.3390/ijerph20176657

**Published:** 2023-08-26

**Authors:** Liliane Pinho de Almeida, Leandro Caetano Guenka, Danielle de Oliveira Felipe, Renato Porfirio Ishii, Pedro Senna de Campos, Thomaz Nogueira Burke

**Affiliations:** 1Allied Health Institute, Federal University of Mato Grosso do Sul, Campo Grande 79070-900, Brazil; lilianepda@ufms.com (L.P.d.A.); danita.felipe@ufms.com (D.d.O.F.); renato.ishii@ufms.br (R.P.I.); pedrosennapsc@ufms.com (P.S.d.C.); thomaz.burke@ufms.br (T.N.B.); 2Medicine, State University of Mato Grosso do Sul, Campo Grande 79115-898, Brazil

**Keywords:** telerehabilitation, motion analysis, capture system

## Abstract

New technologies based on virtual reality and augmented reality offer promising perspectives in an attempt to increase the assessment of human kinematics. The aim of this work was to develop a markerless 3D motion analysis capture system (MOVA3D) and to test it versus Qualisys Track Manager (QTM). A digital camera was used to capture the data, and proprietary software capable of automatically inferring the joint centers in 3D and performing the angular kinematic calculations of interest was developed for such analysis. In the experiment, 10 subjects (22 to 50 years old), 5 men and 5 women, with a body mass index between 18.5 and 29.9 kg/m^2^, performed squatting, hip flexion, and abduction movements, and both systems measured the hip abduction/adduction angle and hip flexion/extension, simultaneously. The mean value of the difference between the QTM system and the MOVA3D system for all frames for each joint angle was analyzed with Pearson’s correlation coefficient (r). The MOVA3D system reached good (above 0.75) or excellent (above 0.90) correlations in 6 out of 8 variables. The average error remained below 12° in only 20 out of 24 variables analyzed. The MOVA3D system is therefore promising for use in telerehabilitation or other applications where this level of error is acceptable. Future studies should continue to validate the MOVA3D as updated versions of their software are developed.

## 1. Introduction

Telerehabilitation or e-rehabilitation is an integral part of e-medicine that allows physiotherapists and patients to control rehabilitation processes from a distance [1]. This form of rehabilitation was developed with the aim of providing the patient and the physiotherapist with greater control over the prescribed therapy so that it can be performed at home [2], reducing the time and cost of hospitalization [3], and allowing an increase in the coverage area of rehabilitation services, especially with respect to reaching patients living long distances from traditional rehabilitation services.

Several studies indicate that telerehabilitation can be useful and as effective as conventional treatments [4,5,6,7]. In addition, telerehabilitation increases adherence, motivation, and frequency of physiotherapy sessions [8]. Another advantage of telerehabilitation systems is the collection of quantitative data related to the therapy and the ease of access and manipulation of these data by health professionals [9,10,11]. Data collected through sensors can be further processed and used to develop more effective interventions [12,13].

There are several different possibilities for applications of motion capture technology. The majority of studies carried out in the field of rehabilitation aim to engage patients in an exercise regime proposed by the physiotherapist. These systems enable patients to receive real-time feedback on the prescribed activity, assist in the correction of joint positions during the execution of the movements, and aid in remote monitoring and adjustment of the exercise prescription between each clinical visit to the patient [14].

Telerehabilitation systems are composed of at least one camera that allows the physiotherapist to see and monitor the patient from a distance (videoconference). More complex systems include sensors capable of motion analysis, which can, in general, be classified into three main groups. The first group comprises systems in which users need to wear devices that capture their movements [15,16,17,18]. The second group includes motion capture systems, which allow users to carry out their activities without the need to wear sensors. Examples of this group are systems that use the Nintendo Wii Remote, Leap Motion, Kinect, or, more recently, depth cameras such as Intel’s RealSense. Finally, the third group seeks to use robotics as a tool for telerehabilitation. Among these systems, we can mention the MOTORE++ [19], HOMEHEAB [20], and the WAM robot [21].

Optical motion analysis techniques have been widely used in biomechanics for measuring large-scale motions, and kinematic data are computed using marker-based motion capture, such as the Qualisys Track Manager system (QTM—Qualisys AB, Gothenburg, Sweden), which is considered as a gold-standard measure for motion assessment [22], albeit expensive and, therefore, restricted to specialized centers, which limits its large-scale dissemination. On the other hand, 2D RGB cameras are widely used in various devices, being present in almost all smartphones sold to the general public; however, they present the ability to deliver variables in only two dimensions (there are no data on depth), thus requiring computational effort to infer 3D variables from 2D variables.

Single-camera markerless motion capture has the potential to facilitate at-home movement assessment due to the ease of setup, portability, and affordable cost of the technology [23]. A human motion capture system with an RGB-D camera or depth camera has a relevant application in research and industry due to its easy use [24]. The development of technology of this nature, if it is shown to be reliable, could represent an alternative for data collection by health professionals, or even for use in telerehabilitation systems, as the end user would need little training, since this technology is widely used.

Therefore, the aim of this work was to develop a monocular and markless 3D motion analysis capture system (MOVA3D) and to compare it against Qualisys Track Manager (QTM).

## 2. Materials and Methods

### 2.1. MOVA3D System

The system consists of an RGB digital camera to capture videos of the movements to be analyzed and software dedicated to the automatic identification of the individual, their joint centers, and body segments (MOVA3D). The computational technique employed is didactically divided into 3 independent stages: (1) automatic detection of human silhouettes in the image, (2) depth referencing between the camera and a reference point on the individual, and (3) calculation of the relative position between joints in the image.

The system generates a data set containing the X, Y, and Z variables of 18 joints or points of interest in the image: “Pelvis”, “R_Hip”, “R_Knee”, “R_Ankle”, “L_Hip”, “L_Knee”, “L_Ankle”, “Torso”, “Neck”, “Nose”, “Head”, “L_Shoulder”, “L_Elbow”, “L_Wrist”, “R_Shoulder”, “R_Elbow”, “R_Wrist”, and “Thorax”. Hip angles were derived in sagittal (flexion/extension) and frontal plane (abduction/adduction) using the 3D position of the “Knee”, “Hip”, and “Thorax” joint centers. “Hip”, Knee”, and “Ankle” were used for knee angle calculations. Range of motion (ROM) detection was performed by creating a motion stop or inversion identifier. The stop identifier algorithm detects, after the start of the movement, if the stop of the movement occurred. This happens by calculating the angular coefficient of the straight line, referring to the angles with variation in time. Thus, when identifying the point where the stability of the slope of the line occurs, the algorithm assumes a stop. In the current study, only the markers that delimit the lower limbs and hip were used in Qualisys analysis.

### 2.2. Subjects and Experimental Design

In the experiment, 10 individuals aged between 22 and 50 years (5 men and 5 women) with a body mass index between 18.5 and 29.9 kg/m^2^ were recruited. A wide range of age and BMI was included in the study to improve the external validity of the results. The exclusion criteria used were the presence of any physical, cognitive, or balance limitation, which could prevent or hinder the execution of the proposed activities during the capture of the movements. The experiment was conducted at the Movement Analysis Laboratory at the Associação de Pais e Amigos dos Excepcionais de Campo Grande (APAE/CG) in Campo Grande/MS, Brazil, from September 2020 to September 2021. Prior to capturing the movements, the individuals answered a questionnaire containing questions on socio-demographic data (email, contact, and age), anthropometric data (weight and height), and their physical condition. The study design was approved by the IRB-Universidade Federal de Mato Grosso do Sul (#2 9358720.7.0000.0021).

The experimental design comprised the concomitant evaluation of the subjects by two systems: Qualisys Track Manager (QTM—Qualisys AB, Sweden) and MOVA3D (own development). The Qualisys system used was configured with seven Oqus series 300, 1280, and 1024 (1.3 pixels) resolution cameras. Figure 1 shows the overall layout. Twenty-six anatomical markers were positioned on anatomical structures, as shown in Table 1. The volume to be collected was previously calibrated following the manufacturer’s recommendations. All assessments were performed by the same professional with experience in the QTM procedures.

The RGB camera of the MOVA3D system (Intel 435i with the depth function disabled) was positioned 1 m from the ground, coinciding with the frontal plane of the subjects. For better centralization of the evaluated individual, the camera was close to the right knee.

#### Experimental Setup and Data Collection

The experiment consisted of each participant performing three groups of activities, 5 repetitions each, while their movements were captured by both systems: squat, hip flexion, and hip abduction. The pace and amplitude of movement were not controlled so that there was naturalness in the execution. Individuals were asked to wear tight clothing (shorts or leggings), and women wore a sports bra, while men were bare-chested. The capture by the MOVA3D system was performed at a frequency of 30 Hz and by the Qualisys system at 100 Hz.

### 2.3. Data Processing

First, the trajectories of the markers with gaps greater than ten frames were interpolated in the data collection software itself (Qualisys Track Manager). The data collected by the Qualisys system were processed using a fourth-order zero-lag Butterworth low-pass filter with a cut-off frequency of 15 Hz, which was previously defined from the cut-off frequency calculation [25]. Since the order number of the filter is determined by the number of poles of its transfer function, the order in this study was then defined according to the complexity of the capacitors and their components [26,27].

With respect to the insertion of the markers, it was observed that each segment of the body needed to have at least 3 markers to ensure subsequent modeling in the 3D software and that, on the contrary, it may generate an overlap and displacement, leading to the need for a more detailed analysis [28]. Therefore, the determination of the points to fix the markers followed the recommendations and the kinematic model of Rocha et al. [29].

Kinematics of the lower limb including landmarks and joint angles assessed with the Qualisys system were calculated in the proximal segment, expressed in relation to Laboratory (global) coordinates with the same orientation. The sequence of rotations used the XYZ-axis, according to the International Society of Biomechanics (ISB) [30]. The range of motion (ROM) was defined as the maximum minus minimum angle during movements in degrees.

The MOVA3D data were processed using a Savitzky–Golay filter with a 17-frame window and a third-degree function. For the alignment of the signals between the two modes of data capture, a cubic interpolation was used, at the beginning of the peaks of the angles evaluated until the last peak, in the same period of time. The recorded data were manually synchronized by the start and end positions of the movement (distinct movement cues) without any interpolation of the data.

The synchronization between the systems was possible based on a specific common temporal event: the fall of an extra marker to the ground at the beginning of the collection of each movement. The following variables were calculated in each frame: knee flexion and extension angle, hip abduction angle, hip flexion, and extension angle.

### 2.4. Correlation of the MOVA3D System with Gold-Standard Measure

For each variable collected, the mean true error and Pearson’s correlation coefficient (r) were calculated. The mean true error was considered as the mean value of the difference between the Qualisys system and the MOVA3D system for all frames, for each joint angle in the analyzed movement. Pearson’s correlation coefficient demonstrates the strength and direction of the relationship between signals. Pearson’s correlation was interpreted according to the guidelines given by [31]: low (less than 0.5), moderate (between 0.5 and 0.75), good (between 0.75 and 0.9), and excellent (above 0.9). The entire data processing routine and statistical analyses were performed in a system developed in Python exclusively for this research.

## 3. Results

The participants, 5 males and 5 females, had a mean age of 30.2 years (95% CI 25.6–34.7), weight of 74.8 kg (95% CI 66.3–83.2), height of 172.4 cm (95% CI 167.2–177.6), and BMI of 25.0 kg/cm^2^ (95% CI 23.1–27.0). Table 2 presents the mean values of the maximum and minimum angles and range of motion (ROM), measured by the two systems in hip abduction, squat, and hip flexion movements. The results are the mean of the 10 individuals evaluated.

It is observed that the range of motion of hip flexion during the squat presented similar results in both assessment systems. Assessments showed no risk to individuals. Device acceptance was excellent.

Table 3 presents the mean error values of the MOVA3D system for the variables maximum angle, minimum angle, and ROM in the hip abduction, squat, and hip flexion exercises. Smaller errors were observed between the maximum and minimum values in hip flexion during the squat and in hip flexion.

Table 4 presents the results of the Pearson correlation index between the Qualysis and MOVA3D systems for the studied variables. Higher correlation coefficients were during right hip abduction and bilateral hip flexion during the squat.

Figure 2 and Figure 3 illustrate, respectively, an example of a subject in the variation in the right hip flexion angle during the hip flexion movement, and the variation in the hip abduction angle during the right hip abduction movement, both in relation to time, measured by the Qualisys system and the MOVA3D system. Note that the mean error remains small throughout the range in Figure 2, and increases towards the end of the movement in Figure 3.

## 4. Discussion

The aim was to compare kinematic measurements of a three-dimensional motion capture and analysis system (MOVA3D), without markers and using a single RGB camera, for use in kinematic analysis. The correlation of this system with the Qualisys system is part of a larger objective of developing technological solutions for use in telerehabilitation systems that allow the physiotherapist to remotely manage and monitor the therapy prescribed at home, as well as increasing adherence to these exercises. This first stage of preliminary data analysis therefore represents an important step towards the construction of low-cost and easy-to-use systems that enable the dissemination of this type of technology in clinical practice. The pros and cons presented facilitate the improvement of these systems and also allow for greater transparency at the time of choice by the clinician.

In general, the MOVA3D system, when correlated to the Qualisys Track Manager (QTM) system, was able to adequately recognize the patterns of hip abduction, squat, and hip flexion movements, with a mean error ranging from 0.55° to 11.10°, except for the maximum angle in right hip abduction and minimum angle in right hip flexion (squat). Other investigations [32] found that most of the parameters have a mean error of 10°, which is comparable to the results obtained through the methodologies illustrated in this work.

As previously mentioned, our results show that the greatest mean true error was in the variable right hip abduction (hip abduction movement), when individuals reach the maximum angle of movement, reaching an error of 41.50°. Despite this, this same variable has a Pearson correlation index of 0.97, indicating that, despite having a high mean error, the variable behaved similarly to that observed by the QTM system (Figure 3).

We suppose that the estimate of the hip joint center made by the MOVA3D system suffers interference during the hip abduction movement, being displaced to the side of the body at the end of the movement, thus decreasing the calculated angle. The error was greater as hip abduction increased. This indicates a possibility of future improvement of the system, with adjustments in the prediction model of hip joint centers.

The variable right hip flexion (squat movement) also showed a high mean true error of 33° at the end of the hip flexion movement (minimum angle of movement). In this case, there was a moderate correlation (r = 0.55) between the MOVA3D and Qualisys measurements. This could have been due to occlusion of the hip by the knee during the end of the squat movement, since a single camera was used (monocular view), and this was positioned in the coronal plane in front of the subjects.

The occlusion of body parts is one of the biggest problems faced by systems without markers [33] that use monocular vision (use of only one camera) for the prediction of joint centers. Movements with large amplitudes performed in an orthogonal plane to the camera, such as hip flexion with the camera positioned in the coronal plane, can cause one joint, such as the knee, to overlap another (hip). In this case, the algorithm is required to predict the joint position without the visual information of the silhouette, which can increase the error.

The majority of studies on this theme in the literature validate systems for kinematic analysis based on depth cameras. These devices have, in addition to the common RGB camera, an infrared sensor, and an infrared point projector. The depth information is calculated through analysis of the projection distortion of the dot pattern by the software built into the camera itself, or by software developed for this purpose. These represent, therefore, real analyses, coming from the capture of direct data.

Our system, in comparison, uses only a standard RGB camera, as seen in cell phones or webcams, with monocular vision (only one lens). This represents almost all cameras built into smartphones sold around the world, which opens up possibilities for using our system on existing mobile devices, without the need to acquire or adapt external devices to capture movements.

Microsoft Kinect has been correlated in several studies for specific squat, hip abduction, and lunge movements, with correlations ranging from 0.18 to 0.83 when correlated to Vicon [32]. During the squat, knee flexion showed a high correlation between the systems (r = 0.88); for hip abduction, values ranged from 0.47 to 0.59; and for lunge, from 0.15 to 0.80 [30]. Schmitz et al. [34] found a correlation of 0.55 between Kinect V2 and Vicon for the evaluation of hip and knee angles during the squat, with results corroborated by Mentiplay et al. [35] for knee flexion during the single-leg squat exercise (r = 0.80). However, when assessing hip abduction, the authors found a low correlation and errors above 15° in the frontal plane. Kotsifaki et al. [36] found high agreement between systems for hip abduction and knee flexion movement during single-leg squat exercises.

Agustsson et al. [37] found a high correlation between the variables from a depth camera attached to an iPad and those collected by the QTM system to assess postural alterations. Vilas Boas et al. [38] found that both versions, V1 and V2, of Microsoft Kinect presented correlations ranging from moderate to poor for hip and upper limb angles during gait analysis, when compared to the Qualisys system. The best associations were observed for the knee joint, ranging from good to excellent [39].

RMS errors for hip and knee joint angles during gait have been reported, ranging from 4° to 10° [40] and 20.15° [38]. In another study, Bahadori et al. [41] found errors of 13.2% (SD 19.6) for knee angle and −3.9% for hip abduction, but with a standard deviation of 75, indicating that the Microsoft Kinect system, when compared to Vicon, is not suitable for clinical use. Tanaka et al. [42] found a moderate to high association between hip flexion and extension angles during gait when compared to the Vicon system.

Other systems such as Capture found a high correlation for heel height variables (r = 0.91) and a low correlation for knee varus (r = 0.29) when correlated to the Vicon system [43].

It can be observed, therefore, that currently, the most widely used systems for the evaluation of human movement are based on the tracking of reflective markers by infrared cameras, as is the case of the Qualisys, Vicon, Optotrak, and Motion Analysis systems. These systems have a high investment value and perform analyses with high precision and accuracy; however, they are difficult to use in clinical evaluations or inappropriate for application in telerehabilitation systems.

Systems without the use of markers (markerless) based on Microsoft Kinect have been developed with the aim of providing a quick and uncomplicated evaluation of human movement, and despite the loss in accuracy and precision, they are capable of evaluating movement patterns satisfactorily in several environments, which may enable their use in clinical and remote monitoring applications. However, despite these characteristics, Microsoft Kinect is not widely accessible to the end user (patients in physiotherapy clinics), and requires the use of specific software and cable connections to a desktop computer. This limits Its use in telerehabilitation systems.

Markerless systems still require refinement in their development, to allow data capture and feedback in real time for the user, with low energy and processing costs, enabling their use in smartphones or tablets. In the near future, the system would be part of telerehabilitation systems, allowing clinicians to assess and follow-up patients at a distance, reducing travels to clinics, and optimizing cost therapeutic performance.

In addition, and of utmost importance, future validation studies in comparison with the gold standard are required that allow more adequate use of these systems in telerehabilitation.

### Limitation

This study has some limitations that affect its application. Firstly, the small number of volunteers did not represent the diversity of body pattern variations necessary to evaluate the capabilities of different anatomical characteristics. The low number of participants was due to COVID-19 restrictions on movement and social distancing.

Furthermore, the signals captured by Qualisys have more than three times the frequency of the RGB camera sensors (100 Hz/30 Hz). Therefore, signals need to be filtered and smoothed more effectively. To achieve this, it is necessary to ensure that the signals are of equal size through interpolation carried out simultaneously with the synchronization process from the beginning of the angular peak estimates to the last peak. A second-order polynomial interpolation function is recommended [44] instead of the cubic interpolation used in this study.

Despite these limitations, the study’s results are an essential step towards the rigorous validation of MOVA3D for clinical assessment of lower limb kinematics. It provides valuable insights into reliable and unreliable kinematic variables.

## 5. Conclusions

The MOVA3D system was shown to be superior to other markerless 3D motion analysis systems, reaching high correlations for six of the eight analyzed variables. The same occurred for the mean error, remaining below 12° in only 4 of the 24 variables analyzed. The MOVA3D system is therefore promising for use in kinematic evaluations with applications in telerehabilitation. The system presents room for improvement in hip abduction and flexion movements.

## Figures and Tables

**Figure 1 ijerph-20-06657-f001:**
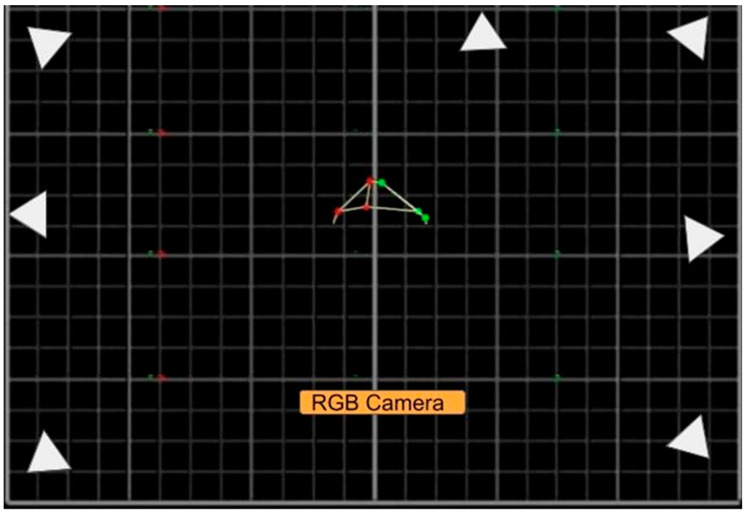
Experimental camera layout for RGB (those labeled in yellow) and QTM (those marked by the white triangles).

**Figure 2 ijerph-20-06657-f002:**
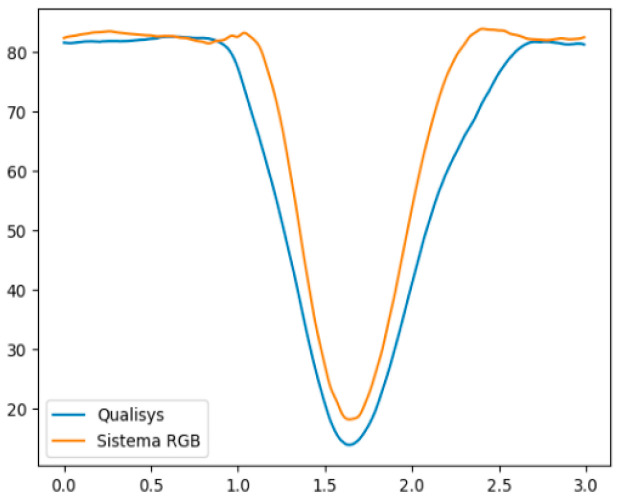
Variation of the right hip flexion angle during the hip flexion movement, measured by the Qualisys system (blue) and the MOVA3D system (orange). The X-axis represents time and the Y-axis the angular variation in degrees.

**Figure 3 ijerph-20-06657-f003:**
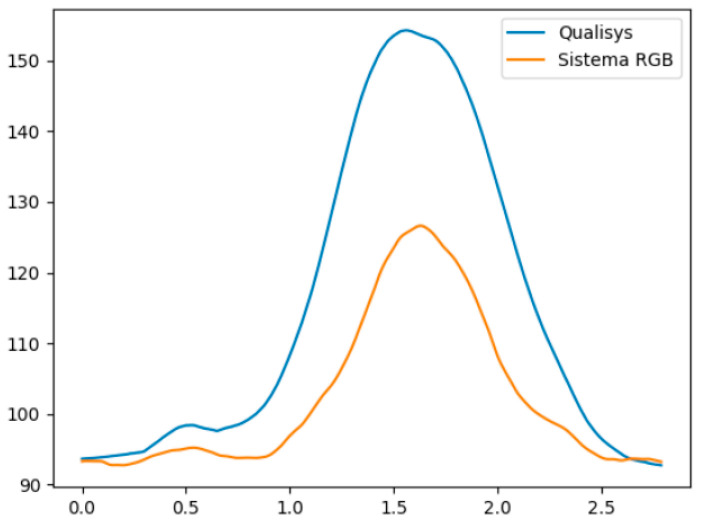
Variation of the right hip abduction angle during the hip abduction movement, measured by the Qualisys system (blue) and the MOVA3D system (orange). The X-axis represents time and the Y-axis the angular variation in degrees.

**Table 1 ijerph-20-06657-t001:** Nomenclature of anatomical reference.

	Nomenclatura	Anatomical Reference
1	R_ASIS	Right Anterior Superior Iliac Spine
2	L_ASIS	Left Anterior Superior Iliac Spine
3	R_PSIS	Right Posterior Superior Iliac Spine
4	L_PSIS	Left Posterior Superior Iliac Spine
5	R_TROC	Right Trochanter
6	L_TROC	Left Trochanter;
7	R_EPIL	Lateral Epicondyle of the Right Femur
8	L_EPIL	Lateral Epicondyle of the Left Femur
9	R_MEPIL	Medial Epicondyle of the Right Femur
10	L_MEPIL	Medial Epicondyle of the Left Femur
11	R_FIBH	Right Fibular Head
12	L_FIBH	Left Fibular Head;
13	R_TTUB	Right Tibial Tuberosity
14	L_TTUB	Left Tibial Tuberosity
15	R_LMAL	Right Lateral Malleolus
16	L_LMAL	Left Lateral Malleolus
17	R_MMAL	Right Medial Malleolus
18	L_LMAL	Left Medial Malleolus
19	R_CAL	Left Calcaneus
20	L_CAL	Right Calcaneus
21	R_1MET	1st Right Metatarsal
22	L_1MET	1st Left Metatarsal
23	R_2MET	2nd Right Metatarsal
24	L_2MET	2nd Left Metatarsal
25	R_5MET	5th Right Metatarsal
26	L-5MET	5th Left Metatarsal

Description of acronyms and respective nomenclatures related to anatomical references.

**Table 2 ijerph-20-06657-t002:** Mean values of the maximum and minimum angles.

Movement		Qualisys			Mova 3D		
		Maximum Angle	Minimum Angle	ROM	Maximum Angle	Minimum Angle	ROM
Hip abduction	ABD_RH	151.5	92.5	59	110	90	20
	ABD_LH	115.9	92.2	23.7	105.2	90.7	14.5
Squat	FLX_RK	65.2	7.5	57.7	54.1	6.9	47.2
	FLX_LK	67.3	7.4	59.9	66.8	10.5	56.5
	FLX_RH	79	24.4	54.3	87.9	57.4	30.6
	FLX_LH	79.6	29.3	50.3	87.3	37.7	49.6
Hip flexion	FLX_RH	81.44	18.44	63	86	21.55	63.11
	FLX_LH	86.66	75.33	11.33	86.11	79.22	6.88

ROM: range of motion; ABD_RH: right hip abduction; ABD_LH: left hip abduction; FLX_RK: right knee flexion; FLX_LK: left knee flexion; FLX_RH: right hip flexion.

**Table 3 ijerph-20-06657-t003:** Mean error of the maximum, minimum, and range of motion.

Movement		Mean Error (Qualisys—Mova 3D)		
		Maximum Angle	Minimum Angle	ROM
Hip abduction	ABD_RH	41.50	2.50	39.00
	ABD_LH	10.70	1.50	9.20
Squat	FLX_RK	11.10	0.60	10.50
	FLX_LK	0.50	−3.10	3.40
	FLX_RH	−8.90	−33.00	23.70
	FLX_LH	−7.70	−8.40	0.70
Hip flexion	FLX_RH	−4.56	−3.11	−0.11
	FLX_LH	0.55	−3.89	4.45

ROM: range of motion; ABD_RH: right hip abduction; ABD_LH: left hip abduction; FLX_RK: right knee flexion; FLX_LK: left knee flexion; FLX_RH: right hip flexion; FLX_LH: left hip flexion.

**Table 4 ijerph-20-06657-t004:** Pearson’s correlation for maximum and minimum.

Pearson’s Correlation					
		r	SD	95% CI	*p*
Hip abduction	ABD_RH	0.97	0.04	0.03	<0.001
	ABD_LH	0.84	0.12	0.07	<0.001
Squat	FLX_RK	0.83	0.17	0.01	<0.001
	FLX_LK	0.94	0.02	0.01	<0.001
	FLX_RH	0.55	0.49	0.03	<0.001
	FLX_LH	0.87	0.05	0.03	<0.001
Hip flexion	FLX_RH	0.93	0.03	0.02	<0.001
	FLX_LH	−0.18	0.65	0.42	<0.001

r: Pearson correlation coefficient; SD standard deviation; 95% CI confidence interval; ABD_RH: right hip abduction; ABD_LH: left hip abduction; FLX_RK: right knee flexion; FLX_LK: left knee flexion; FLX_RH: right hip flexion; FLX_LH: left hip flexion.

## Data Availability

Not applicable.

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
