# Peer review of "Correlation between MOVA3D, a Monocular Movement Analysis System, and Qualisys Track Manager (QTM) during Lower Limb Movements in Healthy Adults: A Preliminary Study"

_ijerph, 2023, doi:10.3390/ijerph20176657_

Round 1
Reviewer 1 Report (New Reviewer)
The MOVA3D system has proven to be viable after important adjustments. The present study should be characterized as "preliminary data".
In the introduction the qualification of the Qualisys Track Manager system (QTM - Qualisys AB, Sweden) has not been clarified. It is the most widely used system in international research; is it reliable? Do several surveys use it as a tool?
The country where the data collection was carried out was not informed.
Moreover, considering that the most restrictive limitations imposed by COVID-19 ceased more than two years ago, it does not seem reasonable to justify the sample size by the pandemic.
The quality of English language is acceptable. It just needs a little revision.
Author Response
"Please see the attachment."

Reviewer 2 Report (New Reviewer)
This paper focuses on developing a markeess 3D motion analysis capture system (i.e., MOVA3D) and at the same time compare it versus Qualisys Track Manager (i.e., QTM)
I suggest authors to rephrase the abstract and make easier to read and also prevent having technical details such as, Intel RealSense camera.
It would be appreciated if authors consider having some visualization for the manuscript to reflect better understanding of the study, e.g., Nomenclature of anatomical references.
Moreover, having a figure representing QTM vs. MOVA3D systems setup as well will be more appreciated.
Author Response
"Please see the attachment."

Reviewer 3 Report (New Reviewer)
Dear Authors,
I congrats with your work. I enclose the reviewed paper along with my comments and suggestions which I hope you may take into consideration.
Good luck and best wishes,

Moderate editing of English language is required
Author Response
"Please see the attachment."

Reviewer 4 Report (Previous Reviewer 1)
Thank you for your manuscript. Your topic is quite interesting; however, I have a major issue regarding Materials and Methods section: the sample size. Moreover, Introduction section must be improved. Please see my comments below…
INTRODUCTION
P2.“Therefore, the aim of this work was to develop a monocular and markless 3D motion analysis capture system (MOVA3D)…” – The methodologies that underlie the construction of the system should be developed in a previous paragraph…
P2.“Therefore, the aim of this work was to develop a monocular and markless 3D motion analysis capture system (MOVA3D) and to concurrently compare it against Qualisys Track Manager (QTM).” – The Qualisys Track Manager (QTM) can be considered a gold standard tool... So, must also be referenced in the previous paragraph…
MATERIALS AND METHODS
P2. “Ten individuals aged between 22 and 50 years (5 men and 5 women) and a body mass index between 18.5 and 29.9 kg/m² were recruited.” The sample is very small and there is a large disparity regarding age and BMI. Moreover, I believe that the analysis should be separated between men and women. These are not the best options to test a markerless system and this is the biggest problem of your study…
Maybe increasing the sample is a solution…
P3. “The RGB camera of the MOVA3D system (Intel 435i with the depth function disabled) was positioned 1 meter from the ground, coinciding with the frontal plane of the subjects.” Why 1 meter? Is this important regarding the results? Which knee is closest to the camera? Left or right? This could influence the results…
In order to validate the instrument, it is necessary to understand whether the data obtained by the 2 methods are similar. And from this point of view it is not enough just to correlate. More statistics is needed...
RESULTS
Why only hip flexion and hip abduction were represented graphically?
Author Response
"Please see the attachment."

Round 2
Reviewer 1 Report (New Reviewer)
The manuscript has been sufficiently improved to warrant publication in IJERPH.
Author Response
Dear reviewer 1,
We appreciate your comments and suggestions.
I hope the manuscript is suitable for publication in its final version.
Best regards
Reviewer 3 Report (New Reviewer)
Dear Authors,
thanks for addressing my comments and suggestions.
Well done.
Author Response
Dear reviewer 3
We appreciate your comments and suggestions.
I hope the manuscript is suitable for publication in its final version.
Best regards
Reviewer 4 Report (Previous Reviewer 1)
The sample is very small and there is a large disparity regarding age and BMI. Moreover, I believe that the analysis should be separated between men and women. These are not the best options to test a markerless system and this is the biggest problem of your study. This is a major concern... Increasing the sample must be one of the solutions, and analysys must consider subjects with similar BMI.
Author Response
Dear reviewer 4
Thanks for your time and dedication reviewing our manuscript. We agree with your suggestions.
The title was changed to enfatize that this is a preliminar study.
A wide range of age and BMI was included in the study to improve the external validity of the results, and this statement was included in section 2.2 (in purple).We do also agree that increasing the sample and and separating it between men and women would significantly improve the paper.
Thanks again for your time. I hope the manuscript is suitable for publication in its final version.
Best regards
This manuscript is a resubmission of an earlier submission. The following is a list of the peer review reports and author responses from that submission.
Round 1
Reviewer 1 Report
Thank you for your manuscript. Your topic is quite interesting; however, I have major issues regarding Materials and Methods section. Moreover, Introduction and Results section must be improved. Please see my comments below…
INTRODUCTION
P1L34. “In addition, telerehabilitation can increase patient adherence and motivation during physiotherapy sessions…” Reference?
P1L34. “In addition, telerehabilitation can increase patient adherence and motivation during physiotherapy sessions, in addition to increasing the frequency of sessions as it is able to be performed at home, with remote supervision by the therapist.” This sentence is quite confusing. Please rephrase.
P2L51. “The second group includes systems using DIT movement…”. It is not explained in the text what “DIT” means.
P2L57. “The majority of solutions for motion assessment, such as the Vicon 3D (Vicon Motion Systems Ltd., Oxford, UK) are expensive and, therefore, restricted to specialized centers, which limits their large-scale dissemination.” I believe that the expression “most solutions” is not the most appropriate expression… I prefer “gold standard measure”, also used later in the text.
P2L67. “Therefore, the aim of the current study was to concurrently validate a 3D motion capture and analysis system, called the MOVA3D, against the gold standard Qualisys Track Manager (QTM).” In line with the previous comment, this sentence suggests that this system is the gold standard measure compared to all others, and that is not true.
P2L69. “The MOVA3D system was previously developed by the research group, and uses a single RGB camera (monocular vision) and dedicated software for inferring 3D kinematic data from 2D images”. This sentence should be developed in a separate paragraph, which should expose the methodologies that underlie the construction of the system.
MATERIALS AND METHODS
P2L84. “With these data it is possible to calculate the variables of clinical interest: absolute angles, and maximum and minimum angles of each joint.” The text never mentions how the angles are calculated or what their values ​​mean; and this is true for both measurements used.
P2L88. “Ten individuals aged between 22 and 50 years (5 men and 5 women) and a body mass index between 18.5 and 29.9 kg/m² were recruited.” The sample is very small and there is a large disparity regarding age and BMI. Moreover, I believe that the analysis should be separated between men and women. These are not the best options to test a markerless system.
P3L102. “The RGB camera of the MOVA3D system (Intel 435i with the depth function disabled) was positioned 1 meter from the ground, coinciding with the frontal plane of the subjects.” Why 1 meter? Is this important regarding the results? Which knee is closest to the camera? Left or right? This could influence the results…
P3L135. “The experiment consisted of each participant performing three groups of activities while their movements were captured by both systems: squat, hip flexion, and hip abduction.” How many trails were performed by each subject regarding each movement?
P3L139. “The capture by the MOVA3D system was performed at a frequency of 30Hz and by the Qualisys system at 100Hz.” Why weren't frequencies that were multiples of each other used? I believe that would have been the best option. See also other comments further ahead.
P3L141. Separate the data processing explanation for the two measurements.
It is not clear why different authors and guidelines were used regarding the placement of the markers…
P4L165. “The MOVA3D data were upsampled to 100Hz by means of a cubic interpolation, so that both systems had an identical sampling rate.” So, the real data were not compared... It doesn't seem right to me… This is a problem related to the chosen frequencies.
P4L165. “The synchronization between the systems was possible based on...” The cubic interpolation and the chosen frequencies do not allow a true synchronization…
In order to validate the instrument, it is necessary to understand whether the data obtained by the 2 methods are similar. And from this point of view it is not enough just to correlate. More statistics is needed...
RESULTS
It is not enough to place the graphs and tables… Results section must be improved…
Table notes are not in line with what is in the table itself
Do the graphs represent the mean values ​​of the various subjects? Is it just an example of a subject? How was it calculated? Reference?
Why only hip flexion and hip abduction were represented graphically?
DISCUSSION
P6L219. “The current study aimed to develop and validate a three-dimensional motion capture and analysis system (MOVA3D)…” Not the best nomenclature…
P6L233. “We believe that, clinically, mean errors of up to 15 degrees are acceptable…” I disagree… Reference?
Responses:
P1L34. “In addition, telerehabilitation can increase patient adherence and motivation during physiotherapy sessions…” Reference?
Reference was added.
P1L34. “In addition, telerehabilitation can increase patient adherence and motivation during physiotherapy sessions, in addition to increasing the frequency of sessions as it is able to be performed at home, with remote supervision by the therapist.” This sentence is quite confusing. Please rephrase.
Paragraph has been rewritten.
P2L51. “The second group includes systems using DIT movement…”. It is not explained in the text what “DIT” means.
Abbreviation was removed and the correct term was described.
P2L57. “The majority of solutions for motion assessment, such as the Vicon 3D (Vicon Motion Systems Ltd., Oxford, UK) are expensive and, therefore, restricted to specialized centers, which limits their large-scale dissemination.” I believe that the expression “most solutions” is not the most appropriate expression… I prefer “gold standard measure”, also used later in the text.
Thank you. The expression was changed.
P2L67. “Therefore, the aim of the current study was to concurrently validate a 3D motion capture and analysis system, called the MOVA3D, against the gold standard Qualisys Track Manager (QTM).” In line with the previous comment, this sentence suggests that this system is the gold standard measure compared to all others, and that is not true.
The sentence was reformulated.
P2L69. “The MOVA3D system was previously developed by the research group, and uses a single RGB camera (monocular vision) and dedicated software for inferring 3D kinematic data from 2D images”. This sentence should be developed in a separate paragraph, which should expose the methodologies that underlie the construction of the system.
The paragraph was restructured.
MATERIALS AND METHODS
P2L84. “With these data it is possible to calculate the variables of clinical interest: absolute angles, and maximum and minimum angles of each joint.” The text never mentions how the angles are calculated or what their values mean; and this is true for both measurements used.
Description of how the calculations were performed has been added on lines 83-88.
P2L88. “Ten individuals aged between 22 and 50 years (5 men and 5 women) and a body mass index between 18.5 and 29.9 kg/m² were recruited.” The sample is very small and there is a large disparity regarding age and BMI. Moreover, I believe that the analysis should be separated between men and women.These are not the best options to test a markerless system.
Unfortunately, the sample has been compromised by the COVID-19 pandemic. In future projects, we expect to increase the sample to be able to perform subgroups analysis matched by age, gender and BMI.
P3L102. “The RGB camera of the MOVA3D system (Intel 435i with the depth function disabled) was positioned 1 meter from the ground, coinciding with the frontal plane of the subjects.” Why 1 meter? Is this important regarding the results? Which knee is closest to the camera? Left or right? This could influence the results…
The camera was positioned 1 meter above the ground to adequately capture the joint centers. The subject was centered, so no knee was closer that other to the camera.
P3L135. “The experiment consisted of each participant performing three groups of activities while their movements were captured by both systems: squat, hip flexion, and hip abduction.” How many trails were performed by each subject regarding each movement?
5 repetitions
P3L139. “The capture by the MOVA3D system was performed at a frequency of 30Hz and by the Qualisys system at 100Hz.” Why weren't frequencies that were multiples of each other used? I believe that would have been the best option. See also other comments further ahead.
We agree with the reviewer. Therefore, the system was developed to be used with regular cameras from smartphones, and 30Hz is the most common frequency in these devices.
P3L141. Separate the data processing explanation for the two measurements. It is not clear why different authors and guidelines were used regarding the placement of the markers…
The paragraph was rewritten and the markers protocol was specified.
P4L165. “The MOVA3D data were upsampled to 100Hz by means of a cubic interpolation, so that both systems had an identical sampling rate.” So, the real data were not compared... It doesn't seem right to me… This is a problem related to the chosen frequencies.
Current research recommends the use of a second-order polynomial interpolation to align and synchronize the signals when they are captured by different frequencies (Qualisys and RGB-D camera). However, data synchronization was performed according to the literature at the beginning of the peaks of the evaluated angles to the last peak, at the same time. We acknowledge the limitations of the present study. These descriptions have been added in the limitations paragraph.
P4L165. “The synchronization between the systems was possible based on...” The cubic interpolation and the chosen frequencies do not allow a true synchronization
The recorded data was manually synchronized by the start and end position of the movement (distinct movement cues) without any interpolation of the data.
In order to validate the instrument, it is necessary to understand whether the data obtained by the 2 methods are similar. And from this point of view it is not enough just to correlate. More statistics is needed...
We agree. The title and the objective were changed.
RESULTS
It is not enough to place the graphs and tables… Results section must be improved…
Results were improved.
Table notes are not in line with what is in the table itself Do the graphs represent the mean values of the various subjects? Is it just an example of a subject? How was it calculated? Reference?
Table notes were all adequate. Graphs is an example of a subject.
Why only hip flexion and hip abduction were represented graphically?
Figures were just an exemplification of a subject's movements. The most complete data were presented in the tables.
DISCUSSION
P6L219. “The current study aimed to develop and validate a three-dimensional motion capture and analysis system (MOVA3D)…” Not the best nomenclature…
The phrase was rewritten.
P6L233. “We believe that, clinically, mean errors of up to 15 degrees are acceptable…” I disagree… Reference?
The value was described wrong. New paragraph with reference was added.
Reviewer 2 Report
Dear Authors,
I appreciate the author's work on the manuscript, however corrections and clarifications should be introduced into the manuscript prior to publication.
Point 1. In case of affiliation with an official institution the official e-mail address should be used.
Point 2. You should provide e-mail addresses for all the co-authors.
Point 3. Please, clarify the moment regarding the name of the university – is it state or federal?
Point 4. Please, correct the multiple mistypes and mistakes, line: 9, 11, 12, 13, 16, 71, 108.
Point 5. All the graphs should be renamed as figures.
Point 6. Please add the Institutional Review Board Statement and approval number as this study does contains human.
Faithfully Yours,
Reviewer
Responses:
Point 1. In case of affiliation with an official institution the official e-mail address should be used.
Ok, but some authors no longer have institutional emails and another is a hired professor.
Point 2. You should provide e-mail addresses for all the co-authors.
Okay, all e-mails have been provided.
Point 3. Please, clarify the moment regarding the name of the university – is it state or federal?
The corresponding author is a professor at the State University of Mato Grosso do Sul and is a postdoc student at the Federal University of Mato Grosso do Sul.
Point 4. Please, correct the multiple mistypes and mistakes, line: 9, 11, 12, 13, 16, 71, 108.
All bugs have been fixed.
Point 5. All the graphs should be renamed as figures.
Changes have been made.
Point 6. Please add the Institutional Review Board Statement and approval number as this study does contains human.
Ethics approval number has been added to the text.